# Oat Hull as a Source of Lignin-Cellulose Complex in Diets Containing Wheat or Barley and Its Effect on Performance and Morphometric Measurements of Gastrointestinal Tract in Broiler Chickens

Tomasz Hikawczuk [1] , Anna Szuba-Trznadel [2] , Patrycja Wróblewska [2,*] and Andrzej Wiliczkiewicz [2]

[1]   Statistical Analysis Centre, Wroclaw Medical University, Karola Marcinkowskiego 2-6, 50-368 Wroclaw, Poland; tomasz.hikawczuk@umw.edu.pl

[2]   Department of Animal Nutrition and Feed Science, Wroclaw University of Environmental and Life Sciences, Chelmonskiego 38c, 61-630 Wroclaw, Poland; anna.szuba-trznadel@upwr.edu.pl (A.S.-T.)

[*]   Correspondence: patrycja.wroblewska@upwr.edu.pl; Tel.: +48-71320-5831

**Abstract:** The purpose of the experiment was to determine the effect of oat hull on the performance and morphometric measurements of the gastrointestinal tract, and to correlate the results of these measurements with the type of the determined dietary fiber in feed and the number of microorganisms. The Asp method is simpler and quicker than non-starch polysaccharide analysis, and can give quick information in the analysis of fiber fractions (soluble and insoluble) in the component or in a diet, and also related the obtained results with the performance of broiler chickens. The utilization of oat hull in the amount of 1% of the diet of broiler chickens results in the highest body weight on the 28th day of life ($p < 0.05$) in comparison to the group not receiving oat hull in the diet and with a 3% share of this structural component. Oat hull in the diet of broiler chickens in the amount of 1% also reduces the total length of the intestines ($p < 0.05$), compared with the share of 0 and 3%. The soluble fiber contained in the grains of barley and wheat has an influence on the higher metabolic weight of the glandular stomach of broiler chickens compared to the birds receiving corn grain in their diet. Barley grain and oat hull in the amount of 3% significantly ($p < 0.01$) increase the weight of gizzards. The increase in the weight of the proventriculus ($r = 0.392$), gizzard ($r = 0.486$) and duodenum ($r = 0.657$) was positively correlated with the growth of *E. coli* bacteria in the crop. The opposite effect in the case of negative correlation was determined in the case of the duodenum and *E. coli* count ($r = -0.593$).

**Keywords:** dietary fiber; oat hull; broiler chickens; performance; morphometric measurements

## 1. Introduction

The classical definition of total dietary fiber describes it as the sum of soluble and insoluble non-starch polysaccharides (NSP) and lignin [1–3]. The nutritional recommendations for broiler chickens take into consideration only a minimum crude fiber content of 3% of the feed [4,5]. The total dietary fiber (TDF) content of individual component soluble (SDF) and insoluble (IDF) fractions are not constant and can affect the performance of broiler chickens [6]. Diets with a high amount of wheat and barley, especially in the case of young broiler chickens (to their 4th week of life), increase the viscosity of chyme, caused by the arabinoxylans or β-glucans present in wheat and barley, which decrease performance, influence changes in the gastrointestinal tract, affect the digestibility of nutrients and impact the microbial status of the gastrointestinal tract [7,8]. One of the solutions to decreasing viscosity in the first 4 weeks of life is the addition of lignin-cellulose complex rich components, such as oat hull, to achieve a diluting effect relative to the SDF fraction [9].

Since the laboratory analyzes commonly used to determine the NSP fraction [10,11] are labor-intensive and time-consuming, other kinds of analysis that are simpler and quicker

are applied. The first one is the Van Soest and Wine analysis [12], which is a faster and less labor-intensive solution that is typically performed when determining the fiber in the feed for ruminants (detergent dietary fiber). Another one is the Asp method [13], which is useful in the analysis of food products [14,15].

Soluble fiber in broiler chickens has the ability to exchange cations in solution, bind and retain water and increase the viscosity of the digesta [16,17]. The excretion of sticky feces by the birds results in an increase in the moisture content of the litter and creates favorable conditions for the development of pathogenic bacteria. The increase in the proportion of these chemical compounds in the diet mainly reduces the digestibility of starch and the total protein [18,19]. The high amount of soluble fiber in the diet increases the length of the jejunum and ileum, which increases the secretory surface of epithelial cells [20]. Moreover, the extended retention time is associated with producing a significant amount of organic acids, lowering the pH, decreasing body weight gains and increasing the feed conversion ratio [8,21,22].

The NSP content in cereal grains and the effect of the increasing viscosity caused by the arabinoxylans and beta glucans can also be related with the total dietary fiber analysis used in human nutrition. The Asp method is simpler and quicker than NSP analysis, and can give quick information in the analysis of fiber fractions (soluble and insoluble) in a component or diet. Of course, it does not provide a more accurate determination than NSP analysis, but it allows for a quick determination of feed components in the case of solubility.

Higher amounts of oat hull in a diet (5–10%) are a source of IDF consisting mainly of the lignin-cellulose complex and its application in the diet, causing the dilution effect which decreases the concentration of nutrients in the case of slow-growing birds and layer hens, which benefits their health [23,24]. On the other hand, small amounts of insoluble fiber (2–3%) can increase the digestibility of starch due to increased amylase activity [25,26]. The addition of a small amount of insoluble fiber can increase the growth rate of birds due to its compensatory effect and also improve their health and welfare due to the decreased risk of necrotic enteritidis and foot dermatitis in broiler chickens [27–29]. Coarse oat hull increased the digesta passage rate compared with finely ground hull and, as a source of insoluble fiber, it increased gizzard weight and size [30,31]. The mutual proportions of the fiber fractions with different solubility in the diet influence the differentiation of organ weights and the length of the intestines in the case of individual sections of the gastrointestinal tract, which can be useful in obtaining offal and reducing the bacterial contamination of meat by decreasing the pH, which will prevent the development of pathogenic bacteria in intestines [32–34].

This research was part of a project funded by the Polish National Science Center related to the use of a small amount of oat hull in the diets of broiler chickens as a source of the lignin-cellulose complex (containing mainly IDF) and its impact on limiting the activity of soluble fiber fractions, which affects the growth rate of birds, differences in the development and microbial status of the gastrointestinal tract and the digestibility of nutrients. The purpose of this experiment was to determine the effect of oat hull on the performance and morphometric measurements of the gastrointestinal tract calculated in terms of metabolic weight, and to determine the correlation between the results of these measurements and the type of the determined dietary fiber, as well as the number of microorganisms in a crop and the proximal part of the ileum.

## 2. Materials and Methods

### 2.1. Animals, Diets and Nutrition

In the experiment, 162 birds were kept in 27 metabolic cages (6 birds in each replication). The design of the experiment used two factors: the source of cereal grain (three: corn, wheat and barley—9 replications in each variant) and the level of oat hull (three: 0, 1 and 3%—9 replications in each variant). The two-way ANOVA with 9 treatments ($3 \times 3$, randomized complete block design) was used. The entire data analysis project was approved by the Local Ethics Committee in Wrocław (protocol No. 084/2010).

The experimental material consisted of day-old broiler chickens of the Hubbard Flex male line. Birds were housed in the same room during the experiment, and grouped in 27 metabolic cages (6 birds in each), to which they were assigned on the basis of similar body weight (about 44.5 g), so as to obtain the effect of homogeneity of variance at the beginning of the experiment and no significant statistical differences, which allowed us to observe differences depending on the type and level of the experimental factor used. Birds were kept until 28 days of age and fed a starter diet (Table 1). In all treatments of the birds, the content of cereal grains (maize, wheat and barley) exceeded 50%, while oat husk was used in the amount from 0 to 3%. As a control diet with a standard composition based on corn and post-extraction soybean meal, the treatment Maize OH 0% was used. All diets were isoenergetic and isoproteic (Table 2) and were prepared according to the Polish Requirements of Poultry Nutrition [4].

**Table 1.** Ingredient composition of diets (%).

| Ingredients | Maize OH ** 0% | Maize OH 1% | Maize OH 3% | Wheat OH 0% | Wheat OH 1% | Wheat OH 3% | Barley OH 0% | Barley OH 1% | Barley OH 3% |
|---|---|---|---|---|---|---|---|---|---|
| Maize | 55.9 | 54.7 | 52.0 | 8.1 | 6.8 | 4.1 | 0.7 | - | - |
| Wheat | - | - | - | 50.0 | 50.0 | 50.0 | - | - | - |
| Barley | - | - | - | - | - | - | 50.0 | 50.0 | 50.0 |
| Soybean meal | 36.9 | 37.0 | 37.4 | 33.4 | 33.6 | 33.9 | 38.2 | 37.8 | 35.8 |
| Soybean oil | 2.8 | 3.0 | 3.3 | 4.1 | 4.3 | 4.7 | 6.9 | 7.0 | 7.0 |
| Oat hull | - | 1.0 | 3.0 | - | 1.0 | 3.0 | - | 1.0 | 3.0 |
| Dicalcium phosphate | 2.11 | 2.00 | 2.03 | 1.93 | 1.85 | 1.85 | 1.72 | 1.72 | 1.72 |
| NaCl | 0.34 | 0.35 | 0.34 | 0.37 | 0.35 | 0.35 | 0.36 | 0.37 | 0.36 |
| Limestone | 0.24 | 0.24 | 0.21 | 0.31 | 0.31 | 0.31 | 0.40 | 0.40 | 0.40 |
| DL- Methionine 98% | 0.21 | 0.21 | 0.22 | 0.23 | 0.22 | 0.22 | 0.22 | 0.22 | 0.22 |
| L-Lysine HCl 78% | - | - | - | 0.07 | 0.07 | 0.07 | - | - | - |
| $Cr_2O_3$ | 0.5 | 0.5 | 0.5 | 0.5 | 0.5 | 0.5 | 0.5 | 0.5 | 0.5 |
| Premix DKA-s * | 1.0 | 1.0 | 1.0 | 1.0 | 1.0 | 1.0 | 1.0 | 1.0 | 1.0 |

* Provided the following per kilogram of diet: vitamin A 10,000 IU, vitamin $D_3$ 2000 IU, vitamin E 20 mg, vitamin K 3 mg, vitamin $B_1$ 2.5 mg, vitamin $B_6$ 0.4 mg, vitamin $B_{12}$ 0.015 mg, nicotinic acid 60 mg, pantothenic acid 8 mg, folic acid 1.2 mg, choline chloride 450 mg, DL-Methionine 1.0 mg, Mn 74 mg, Fe 30 mg, Zn 45 mg, Cu 4 mg, Co 0.4 mg, I 0.3 mg, ** OH—oat hull.

Broiler chickens were kept under standard rearing conditions. The initial temperature on the first day of the experiment was 32 °C, and it was systematically lowered to 24 °C in the second week of the experiment, 21 °C on the 21st day of the birds' lives and 20 °C on the last day of the experiment.

The lighting program also changed over time from 22 h of light and 2 h of darkness per day in the first seven days, and then from the 7th day onwards, the time that the light was kept on was successively reduced to 18 h and 6 h of darkness per day. The humidity in the experimental chicken house ranged from 61 to 72%. Feed was offered to birds in a loose form semi ad libitum to feeders suspended on cages, and was available by 12 h every day. Water was taken by chickens from nipple system drinkers (their heights were regulated depending on the age of the chickens). During the experiment, the amount of unused feed from the feeders was noted.

**Table 2.** Chemical composition of experimental diets.

| Specification | Maize OH ** 0% | Maize OH 1% | Maize OH 3% | Wheat OH 0% | Wheat OH 1% | Wheat OH 3% | Barley OH 0% | Barley OH 1% | Barley OH 3% |
|---|---|---|---|---|---|---|---|---|---|
| Metabolic energy (MJ·kg$^{-1}$) | 12.2 | 12.3 | 12.2 | 12.2 | 12.2 | 12.2 | 12.2 | 12.2 | 12.2 |
| **Nutrients (%)** | | | | | | | | | |
| Dry mater | 92.7 | 92.2 | 91.9 | 91.6 | 90.0 | 90.1 | 90.1 | 92.0 | 90.1 |
| Crude protein | 22.5 | 22.5 | 21.8 | 21.2 | 21.1 | 21.3 | 22.6 | 23.1 | 21.8 |
| Ether extract | 4.93 | 5.28 | 5.49 | 5.32 | 5.71 | 5.84 | 8.38 | 8.42 | 7.80 |
| Crude ash | 4.11 | 4.08 | 4.10 | 3.87 | 3.93 | 4.02 | 5.07 | 5.12 | 5.09 |
| **Structural components (%)** | | | | | | | | | |
| Crude fiber | 2.94 | 3.12 | 3.67 | 3.20 | 3.51 | 4.07 | 4.81 | 5.04 | 5.45 |
| TDF * | 20.3 | 21.2 | 22.0 | 19.2 | 19.9 | 20.9 | 24.8 | 25.3 | 26.3 |
| SDF * | 1.21 | 1.21 | 1.22 | 2.58 | 2.61 | 2.63 | 1.70 | 1.74 | 1.69 |
| IDF * | 19.1 | 20.0 | 20.8 | 16.6 | 17.3 | 18.3 | 23.1 | 23.6 | 24.7 |
| NDF * | 11.0 | 11.5 | 12.6 | 15.9 | 16.1 | 17.6 | 15.1 | 15.6 | 16.1 |
| ADF * | 3.84 | 4.24 | 4.71 | 4.12 | 4.43 | 5.01 | 5.15 | 5.55 | 6.15 |
| Hemicelluloses | 7.15 | 7.21 | 7.86 | 11.8 | 11.7 | 12.6 | 9.96 | 10.1 | 10.7 |
| **Count of microorganisms in feed (log CFU·g$^{-1}$)** | | | | | | | | | |
| TAMC * | 1.45 | 1.44 | 1.40 | 0.34 | 0.33 | 0.29 | 1.29 | 1.28 | 1.30 |
| *Lactobacillus* spp. | 2.13 | 2.12 | 2.11 | 0.89 | 0.89 | 0.87 | 1.26 | 1.26 | 1.28 |
| TYMC * | 1.91 | 1.90 | 1.86 | 1.82 | 1.80 | 1.76 | 1.46 | 1.44 | 1.43 |
| *E. coli* | - | - | - | - | - | - | - | - | - |
| *Salmonella* sp. | - | - | - | - | - | - | - | - | - |

* TDF—total dietary fiber, SDF—soluble dietary fiber, IDF—insoluble dietary fiber, NDF—neutral dietary fiber, ADF—acid dietary fiber, TAMC—total aerobic microbial count, TYMC—total yeast and mold count, OH—oat hull amount, ** OH—oat hull.

### 2.2. Performance of Broiler Chickens

During the experiment, the performance of broiler chickens was noted in the form of their body weight and feed intake, and on this basis, the feed conversion ratio was calculated. Animals were weighed at 7, 11, 14, 18, 21, 25 and 28 days of age, and the feed intake amount was also determined on those days. Additionally, the mortality of chickens in individual replications was recorded across the whole experiment, but it did not exceed 3% in any of the treatments; therefore, it was not included in the results presented in the tables.

### 2.3. Dissection of Organs from the Upper Part of GIT

On day 28 of the experiment, 81 chickens were sacrificed by concussion following a truncheon strike and a quick sublingual incision for the bleeding of birds. Then, selected organs or sections of intestines were dissected from the gastrointestinal tract. From the upper part of the gastrointestinal tract (GIT), the empty crop, proventriculus and gizzard were dissected and weighed.

During the dissection, individual sections of the intestines were also measured: the duodenum after separation from the pancreas, the jejunum to Meckel's diverticulum, the ileum from Meckel's diverticulum and the ceca and large intestine. In addition, the values corresponding to the length of the intestines were summed and their total length was determined, which then made it possible to determine the percentage share of each section in the total length of the intestine. For the calculations, all the values related to the organs and intestinal sections were converted to metabolic weight (BW$^{0.67}$).

### 2.4. Chemical Analysis of the Content of Nutrients and Structural Components in Feed

The metabolic energy in individual diets was calculated as the sum of the energy content of individual feed components. For this purpose, the European Tables of Energy

Values for Poultry [35] were used. The Polish Requirements of Poultry Nutrition were used to determine the required level of metabolic energy and nutrients for starter diets [4]. Each diet in each treatment contained in 1 kg of feed 12.5 MJ of metabolic energy and about 22% of crude protein (Table 2).

The chemical composition of the feed ingredients and diets were determined according to the Official Methods of Analysis (AOAC) [36]. Dry matter (DM, AOAC; 934.01); crude protein (CP, Kjeldahl method, AOAC 984.13); crude ash (CA, AOAC 942.05), ether extract (EE, Soxhlet method, AOAC, 920.39A), crude fiber (CF, Hennenberg and Stohmann method, AOAC 978.10); neutral detergent fiber (NDF, Van Soest method, AOAC 2002:04); acid detergent fiber (ADF, Van Soest method, AOAC 973.18); total dietary fiber (TDF, Asp method, AOAC 991.43); and soluble and insoluble dietary fiber (SDF and IDF, Asp method, AOAC 991.42/43). Hemicelluloses were calculated as the difference between NDF and ADF.

### 2.5. Microbial Analysis

The microorganisms in particular sections of the bird's gastrointestinal tract and in the feed were determined in 1 g of the food taken from the crop, the initial part of the ileum and in the feed. To the samples, 9 mL of buffered peptone water was added. In the second step, the sample was homogenized in the Lab-Blender 400 apparatus for 2 min, then successive decimal dilutions were made, and after that, inoculations were made on or into the growth medium, according to the methods described by Wróblewska et al. [37].

### 2.6. Statistical Analysis

The numerical data obtained within the replications were calculated first as a mean value for birds in one replication in the treatment (outliers were brought to the replication mean) using the Excel program. In the second step, all values from the 9 replications in each treatment were evaluated statistically by the two-way ANOVA using the Tibco Statistica 13.3 program [38]. The following experimental model was used:

$$y_{ijk} = \mu + \alpha_i + \beta_j + (\alpha\beta)_{ij} + \varepsilon_{ijk}$$

$y_{ijk}$—value of observed dependent variable
$\mu$—general mean in population (effect of common factors)
$\alpha_i$—effect of cereal
$\beta_j$—effect of oat hull
$(\alpha\beta)_{ij}$—interaction between cereal and oat hull
$\varepsilon_{ijk}$—effect of random factors

Additionally, in tables instead of standard deviation, the standard error of measurement (SEM) was used. In the case of morphometric measurements, the two-way ANCOVA design was used, composed of a one covariate with a two-way ANOVA factorial design, which helps to make the results of the morphometric measurements less dependent on the differences in the final body weight of the broiler chickens. The following experimental model was used:

$$y_{ijk} = \mu + \alpha_i + \beta_j + \gamma\left(x_{ijk} - \overline{x}\right) + \varepsilon_{ijk}$$

$y_{ijk}$—value of observed dependent variable,
$\mu$—general mean in population (effect of common factors)
$\alpha_i$—effect of cereal
$\beta_j$—effect of oat hull
$\gamma(x_{ijk} - \overline{x})$—value of covariate corresponding to variable $y_{ijk}$
$\varepsilon_{ijk}$—effect of random factors

With regard to the data in treatments, the homogeneity of variance (Levene's test) and normality distribution (Saphiro-Wilk's test) were checked. Logarithmic transformation was used when considering the number of microorganisms in the feed, crop and proximal part of the ileum. Duncan's test was used to determine the differences of the mean values between treatments, and were evaluated on two levels of statistical significance at $p < 0.05$

and $p < 0.01$. In the case of correlation, the Pearson test was used and significant coefficients were taken into consideration during the description.

## 3. Results

### 3.1. Performance of Broiler Chickens

At 7 days of age, the highest body weight ($p < 0.01$, Table 3) was shown by chickens fed a maize-based diet (135.2 g) compared to birds fed a barley-based diet (126.9 g), while the use of oat hull (in mean 130.0 g) decreased body weight as compared to the lack of it in the diet of birds (133.6 g). On day 11, the birds receiving corn diets (252.2 g) were still the heaviest ($p < 0.01$) compared to those receiving barley diets (234.9 g). At 14 days of age, a significant difference in body weight ($p < 0.01$) was found between the chickens fed diets with maize and wheat (361.8 g and 354.5 g, respectively) and those fed the barley diets (339.4 g). At that time, a higher body weight was also noted ($p < 0.01$) in the case of chickens fed the diet containing 3% of oat hull (358.5 g) compared to 1% of this component in the diet (343.2 g). On day 28, the highest body weight ($p < 0.01$) was achieved by the birds receiving the barley grain diets (1237 g) compared to the wheat-based formulas (1172 g). On the other hand, the use of 1% of oat hull increased ($p < 0.01$) the body weight of chickens in this period (1240 g) in comparison with the share of 3% of the amount of this component in diets (1184 g).

**Table 3.** Body weight of broiler chickens.

| Specification | Body Weight (g) | | | | | | |
| --- | --- | --- | --- | --- | --- | --- | --- |
| | **Day 7** | **Day 11** | **Day 14** | **Day 18** | **Day 21** | **Day 25** | **Day 28** |
| **Cereal** | | | | | | | |
| Corn | 135.2 [A] | 252.2 [Aa] | 361.8 [A] | 550.9 | 728.8 | 1004.3 | 1216.3 [a] |
| Wheat | 131.5 [a] | 241.1 [b] | 354.5 [A] | 540.7 | 707.2 | 991.2 | 1172.4 [Bb] |
| Barley | 126.9 [Bb] | 234.9 [B] | 339.4 [B] | 535.6 | 712.8 | 1011.0 | 1237.5 [Aa] |
| **Oat hull** | | | | | | | |
| 0% | 133.6 [a] | 244.3 | 353.9 [a] | 549.5 | 718.6 | 985.4 | 1201.9 [ab] |
| 1% | 129.8 [b] | 239.2 | 343.2 [Bb] | 530.8 | 713.4 | 1022.8 | 1240.1 [a] |
| 3% | 130.2 [b] | 244.7 | 358.5 [A] | 547.0 | 716.7 | 998.4 | 1184.1 [b] |
| **SEM** | 1.043 | 1.896 | 2.600 | 3.749 | 4.735 | 8.611 | 11.586 |
| ***p*-value** | | | | | | | |
| Cereal | 0.000 | 0.000 | 0.000 | 0.224 | 0.109 | 0.536 | 0.011 |
| Oat hull | 0.015 | 0.180 | 0.003 | 0.089 | 0.873 | 0.128 | 0.030 |
| Cereal * Oat hull | 0.001 | 0.521 | 0.881 | 0.625 | 0.054 | 0.036 | 0.014 |

* Means in the same row with different superscripts a, b are significantly different with $p \leq 0.05$ and A, B with $p \leq 0.01$.

In the period from day 1 to day 7, significant differences ($p < 0.01$) were found in the feed intake between chickens receiving 0 and 1% of oat hull in their diet (22.1 and 22.3 g·day$^{-1}$) compared to the amount of 3% of oat hull in their diet (21.0 g·day$^{-1}$, Table 4). From the 7th to the 11th day, a lower feed intake ($p < 0.01$) was recorded in the group where the main constituent was barley (28.6 g·day$^{-1}$) compared to chickens fed with wheat or corn diets (30.1 and 30.3 g·day$^{-1}$). However, in the same period, the feed with a 3% share of oat hull (30.5 g·day$^{-1}$) was consumed to a greater extent ($p < 0.01$) than those with a 0 or 1% amount of oat hull. From the 11th to the 14th day, a significantly higher ($p < 0.05$) feed intake of the wheat diets (35.2 g·day$^{-1}$) was found compared to the diets which were based on barley grain 33.6 (g·day$^{-1}$).

**Table 4.** Feed intake of broiler chickens.

| Specification | Feed Intake (g/head/day) | | | | | | |
|---|---|---|---|---|---|---|---|
| | Day 7 | Day 11 | Day 14 | Day 18 | Day 21 | Day 25 | Day 28 |
| **Cereal** | | | | | | | |
| Corn | 21.4 | 30.3 [A] | 34.6 [ab] | 42.6 | 50.1 | 58.7 | 64.5 [a] |
| Wheat | 22.2 | 30.1 [A] | 35.2 [a] | 43.1 | 49.2 | 58.9 | 63.8 [A] |
| Barley | 21.6 | 28.6 [B] | 33.6 [b] | 42.4 | 49.7 | 60.7 | 67.3 [Bb] |
| **Oat hull** | | | | | | | |
| 0% | 22.1 [A] | 28.9 [A] | 34.1 | 42.4 | 49.1 | 57.9 [B] | 64.3 |
| 1% | 22.3 [A] | 29.6 [AB] | 34.3 | 42.3 | 50.3 | 61.1 [A] | 66.7 |
| 3% | 21.0 [B] | 30.5 [B] | 35.1 | 43.4 | 49.6 | 59.3 [AB] | 64.5 |
| **SEM** | 0.273 | 0.363 | 0.316 | 0.229 | 0.248 | 0.552 | 0.599 |
| ***p*-value** | | | | | | | |
| Cereal | 0.084 | 0.006 | 0.033 | 0.365 | 0.279 | 0.108 | 0.016 |
| Oat hull | 0.003 | 0.013 | 0.182 | 0.097 | 0.111 | 0.016 | 0.099 |
| Cereal * Oat hull | 0.000 | 0.000 | 0.007 | 0.245 | 0.185 | 0.018 | 0.165 |

\* Means in the same row with different superscripts a, b are significantly different with $p \leq 0.05$ and A, B with $p \leq 0.01$.

From day 21 to day 25, a significant increase in feed intake ($p < 0.05$) was found in the case of chickens fed diets with a 1% share of oat hull (57.9 g·day$^{-1}$) compared to the birds not receiving this component in their diets (61.1 g·day$^{-1}$). Until day 28, no statistically significant differences ($p < 0.05$) were found in the case of feed intake depending on the amount of oat hull in diets. On the other hand, higher feed intake ($p < 0.01$) was shown by chickens fed barley diets (67.3 g·day$^{-1}$) compared to wheat-based formulas (63.8 g·day$^{-1}$).

From day 1 to day 7 (Table 5), feed conversion per kg body weight gain ($p < 0.01$) was significantly higher for chickens fed wheat and barley grain diets (1.18 and 1.19 kg of feed·kg ADG$^{-1}$, respectively). In the same period, the application of 1% of oat hull in the diet increased ($p < 0.01$) feed conversion (1.20 kg of feed·kg ADG$^{-1}$) compared to a 0 or 3% amount of this component in diets (respectively, 1.16 and 1.13 kg of feed·kg ADG$^{-1}$, Table 5).

**Table 5.** Feed conversion of broiler chickens.

| Specification | Feed Conversion (kg feed·kg ADG$^{-1}$) | | | | | | |
|---|---|---|---|---|---|---|---|
| | Day 7 | Day 11 | Day 14 | Day 18 | Day 21 | Day 25 | Day 28 |
| **Cereal** | | | | | | | |
| Corn | 1.11 [A] | 1.32 [A] | 1.34 | 1.39 | 1.44 | 1.46 | 1.49 |
| Wheat | 1.18 [B] | 1.37 [Bb] | 1.39 | 1.44 | 1.46 | 1.49 | 1.52 |
| Barley | 1.19 [B] | 1.34 [a] | 1.39 | 1.43 | 1.46 | 1.50 | 1.52 |
| **Oat hull** | | | | | | | |
| 0% | 1.16 [Ab] | 1.30 [A] | 1.35 | 1.39 [a] | 1.43 | 1.47 | 1.50 |
| 1% | 1.20 [B] | 1.36 [B] | 1.40 | 1.44 [b] | 1.48 | 1.49 | 1.51 |
| 3% | 1.13 [Aa] | 1.37 [B] | 1.37 | 1.43 [b] | 1.45 | 1.49 | 1.53 |
| **SEM** | 0.013 | 0.015 | 0.013 | 0.009 | 0.008 | 0.011 | 0.012 |
| ***p*-value** | | | | | | | |
| Cereal | 0.000 | 0.006 | 0.088 | 0.074 | 0.486 | 0.233 | 0.236 |
| Oat hull | 0.000 | 0.000 | 0.153 | 0.046 | 0.056 | 0.534 | 0.581 |
| Cereal * Oat hull | 0.000 | 0.000 | 0.024 | 0.076 | 0.299 | 0.016 | 0.023 |

\* Means in the same row with different superscripts a, b are significantly different with $p \leq 0.05$ and A, B with $p \leq 0.01$.

From day 1 to day 11, the highest feed conversion ($p < 0.01$) was found in the group of chickens fed a diet with wheat grain (1.37 kg·kg ADG$^{-1}$) compared to the chickens fed

a corn grain diet (1.32 kg·kg ADG$^{-1}$). Oat hull in the amount of 1 and 3% of diets at that time significantly increased ($p < 0.01$) feed conversion (respectively, 1.36 and 1.37 kg·kg ADG$^{-1}$). In the period from 1 to 18 days of age, the use of oat hull in a proportion of 1 and 3% increased ($p < 0.05$) feed conversion (1.44 and 1.43 kg of feed·kg ADG$^{-1}$, respectively).

### 3.2. The Development of Gastrointestinal Tract

There was no effect of experimental factors ($p < 0.05$) on the metabolic weight of the crop without content (Table 6). A heavier glandular stomach ($p < 0.01$) was observed for chickens receiving barley grain in their diets (3.6 g) compared to birds receiving corn (3.3 g). There was also a statistically significant difference ($p < 0.01$) in the metabolic weight of gizzards between the chickens fed diets containing barley grain (9.6 g) and the birds fed corn or wheat feeds (8.7 g). The use of 3% of oat hull increased ($p < 0.01$) the metabolic weight of the gizzard (9.6 g) in broiler chickens compared to the groups without this component in their diets (8.4 g). The barley grain diets also increased ($p < 0.01$) the duodenal length in broiler chickens (8.8 g) compared to the maize and wheat treatments (8.2 and 8.4 g, respectively).

**Table 6.** Metabolic weight of organs and lengths of intestines recalculated for metabolic weight (BW$^{0.67}$) in broiler chickens ($n = 27$).

| Specification | Slaughter Weight (kg) * | Weight (g) | | Length (cm) | | | | | |
|---|---|---|---|---|---|---|---|---|---|
| | | Crop without Digesta | Proventriculus | Gizzard | Duodenum | Jejunum | Ileum | Large Intestine | Ceca (sum) |
| **Cereal** | | | | | | | | | |
| Corn | 1.22 | 3.1 | 3.3 [Bb] | 8.7 [B] | 8.2 [B] | 15.8 | 15.8 | 3.2 | 9.5 |
| Wheat | 1.25 | 3.3 | 3.4 [a] | 8.7 [B] | 8.4 [B] | 16.2 | 16.1 | 3.0 | 9.9 |
| Barley | 1.31 | 3.5 | 3.6 [A] | 9.6 [A] | 8.8 [A] | 16.1 | 16.1 | 3.3 | 10.0 |
| **Oat hull** | | | | | | | | | |
| 0% | 1.26 | 3.6 | 3.3 | 8.4 [B] | 8.5 | 16.3 | 16.3 [A] | 3.1 | 9.8 |
| 1% | 1.30 | 3.1 | 3.5 | 8.9 [AB] | 8.4 | 15.8 | 15.6 [Bb] | 3.2 | 9.6 |
| 3% | 1.25 | 3.2 | 3.4 | 9.6 [A] | 8.6 | 16.0 | 16.1 [a] | 3.2 | 10.0 |
| **SEM** | 0.012 | 0.110 | 0.042 | 0.153 | 0.089 | 0.102 | 0.105 | 0.044 | 0.111 |
| ***p*-value** | | | | | | | | | |
| Cereal | - | 0.769 | 0.006 | 0.003 | 0.001 | 0.101 | 0.267 | 0.613 | 0.085 |
| Oat hull | - | 0.179 | 0.180 | 0.000 | 0.781 | 0.204 | 0.019 | 0.114 | 0.489 |
| Slaughter weight ** | - | 0.204 | 0.508 | 0.779 | 0.125 | 0.160 | 0.334 | 0.825 | 0.113 |

\* Means in the same row with different superscripts a, b are significantly different with $p \leq 0.05$ and A, B with $p \leq 0.01$, ** covariate.

The exception is the ileum, where the 1% amount of oat hull in the diets decreased ($p < 0.05$) its length in correlation with the slaughter weight (15.6 cm) compared to the lack of this component in the diet (16.3 cm). The use of different cereal grains did not have a statistically significant ($p > 0.05$) effect on the total intestinal length in terms of metabolic weight (Table 7). On the other hand, the use of 1% of oat hull in the diet of chickens lowered ($p < 0.05$) the total intestinal length of intestines (52.6 cm) compared to 0 and 3% of this component in the diet (54.0 and 53.8 cm).

The barley grain in diets for broiler chickens increased ($p < 0.05$) the percentage of duodenum in the total intestinal length (16.2%) compared to birds receiving corn or wheat as a main component (15.7 and 15.6%, respectively).

Moreover, a lower percentage share of jejunum ($p < 0.01$) in broiler chickens in the total intestinal length was observed when barley grain was used in the diets (29.6%) compared to birds that received feeds containing corn or wheat (by 30.1% and 30.0%). The use of 1% of oat hull in the diets also reduced ($p < 0.01$) the share of the jejunum in the total intestinal length (29.6%) compared to the feeds where 0 and 3% of oat hull was used (30.1% and 29.9%).

**Table 7.** Percentage share of total length of intestines recalculated for metabolic weight ($BW^{0.67}$) in broiler chickens.

| Specification | Slaughter Weight (kg) * | Total Length of Intestines (cm) | Percentage Share of Total Length of Intestines (%) | | | | |
|---|---|---|---|---|---|---|---|
| | | | Duodenum | Jejunum | Ileum | Large Intestine | Ceca |
| **Cereal** | | | | | | | |
| Corn | 1.22 | 52.5 | 15.7 [b] | 30.1 | 30.1 [A] | 6.0 | 18.2 |
| Wheat | 1.25 | 53.6 | 15.6 [b] | 30.2 | 30.0 [a] | 5.7 | 18.5 |
| Barley | 1.31 | 54.3 | 16.2 [a] | 29.7 | 29.6 [Bb] | 6.0 | 18.4 |
| **Oat hull** | | | | | | | |
| 0% | 1.26 | 54.0 [a] | 15.7 | 30.2 | 30.1 [A] | 5.8 | 18.2 |
| 1% | 1.30 | 52.6 [b] | 16.0 | 30.1 | 29.6 [Bb] | 6.0 | 18.3 |
| 3% | 1.25 | 53.8 [a] | 15.9 | 29.7 | 29.9 [a] | 5.9 | 18.5 |
| **SEM** | 0.012 | 0.327 | 0.109 | 0.107 | 0.084 | 0.073 | 0.144 |
| *p*-**value** | | | | | | | |
| Cereal | - | 0.663 | 0.016 | 0.087 | 0.032 | 0.102 | 0.783 |
| Oat hull | - | 0.034 | 0.326 | 0.161 | 0.008 | 0.480 | 0.603 |
| Slaughter weight ** | - | 0.014 | 0.154 | 0.288 | 0.694 | 0.660 | 0.784 |

* Means in the same row with different superscripts a, b are significantly different with $p \leq 0.05$ and A, B with $p \leq 0.01$; ** covariate.

### 3.3. Relation between Development of Gastrointestinal Tract and Microorganisms

A medium positive correlation ($p < 0.05$) was found for the weight of the proventriculus and gizzard versus the count of *E. coli* in the crop (r = 0.392 and r = 0.486, respectively, Table 8). On the other hand, a strong positive correlation ($p < 0.05$) was confirmed between the length of the duodenum and the number of *E. coli* in the crop (r = 0.657). In the case of other dependencies, no statistically significant correlations ($p > 0.05$) were found.

**Table 8.** Correlation between weights of organs and lengths of sections of intestines and count of microorganisms in crop and ileum [37].

| Specification | TAMC | *E. coli* | TYMC | *Lactobacillus* spp. |
|---|---|---|---|---|
| **Crop** | | | | |
| Crop without digesta | 0.052 | −0.040 | 0.129 | −0.078 |
| Proventriculus | 0.187 | 0.392 * | −0.173 | 0.217 |
| Gizzard | −0.057 | 0.486 * | −0.204 | −0.119 |
| Duodenum | 0.043 | 0.657 * | −0.171 | 0.184 |
| Jejunum | 0.203 | 0.160 | 0.215 | −0.266 |
| Ileum | 0.133 | 0.169 | −0.034 | −0.035 |
| Large intestine | −0.351 | 0.131 | −0.035 | −0.367 |
| Ceca | 0.213 | 0.207 | −0.125 | 0.119 |
| **Ileum** | | | | |
| Crop without digesta | 0.114 | 0.013 | −0.526 * | 0.167 |
| Proventriculus | −0.079 | −0.236 | 0.041 | −0.393 * |
| Gizzard | 0.346 | −0.017 | 0.123 | −0.247 |
| Duodenum | −0.114 | −0.593 * | 0.191 | −0.465 * |
| Jejunum | 0.407 * | 0.017 | −0.213 | −0.385 * |
| Ileum | 0.177 | −0.275 | −0.212 | −0.330 |
| Large intestine | 0.349 | −0.003 | 0.045 | 0.037 |
| Ceca | 0.096 | −0.114 | −0.164 | −0.289 |

* Significant correlation *p*-value < 0.05.

A strong negative correlation ($p < 0.05$) was found between the weight of crop without digesta and TYMC (r = −0.526, Table 8). Between the weight of the proventriculus

and the number of *Lactobacillus* spp., a medium negative correlation was found ($p < 0.05$; $r = -0.393$).

Duodenal length was strongly negatively correlated with the number of *E. coli* ($p < 0.05$; $r = -0.593$) and medium negatively correlated with the number of *Lactobacillus* spp. ($p < 0.05$; $r = -0.465$). The length of the jejunum was medium positively correlated ($p < 0.05$) with TAMC ($r = 0.407$) and medium negatively correlated with the number of *Lactobacillus* spp.

### 3.4. Relation between Development of Gastrointestinal Tract and Dietary Fiber Fractions

A strong positive correlation ($p < 0.01$, Table 8) was determined with respect to proventriculus weight and crude fiber content, TDF and ADF ($r = 0.586$; $r = 0.502$ and $r = 0.552$, respectively).

The weight of gizzard was strongly positively correlated with the crude fiber content ($r = 0.733$), TDF ($r = 0.662$), IDF ($r = 0.607$) and ADF ($r = 0.785$). In the case of duodenum length, a strong correlation was found with the crude fiber content in diets ($r = 0.666$). While medium for TDF ($r = 0.599$), IDF ($r = 0.533$) and ADF ($r = 0.575$).

## 4. Discussion

### 4.1. Performance of Broiler Chickens

Analysis of the body weight of the broiler chickens in the experiment shows that birds achieved the highest body weight in the group receiving maize grain, and the least body weight in the group receiving wheat grain diets. On the other hand, a slight increase in the share of oat hull in the diet did not affect the intake and conversion of feed, which has been confirmed by the studies by Adibmoradi et al. [28], who found better digestibility of starch and crude protein with a small addition of oat hull to the diet.

It depends mainly on the age of the birds and the type of component that is the source of the soluble and insoluble structural carbohydrates used in the concentrated mixture, which are the building blocks of cell walls [39–41]. Mulla et al. [42], using a 2% share of oat hull in the diet, noted a slightly higher body weight of broiler chickens at 28 days of age (1371 g) compared to the control group receiving concentrated mixtures with corn and soybean extraction (1366 g). On the other hand, Adewole [43] reports that extruding oat hull does not ($p > 0.05$) affect body weight gain during the 36-day rearing period of broiler chickens (2332 g), compared to non-extruded hull (2308 g) and the control group (2449 g).

The factor that limited feed intake in the first days of bird life was too high a level of fiber in the diets based mainly on barley (over 5%). In the later period of bird growth, the highest feed intake was recorded in the groups of chickens fed barley grain diets. At the same time, the addition of oat hull did not affect the digestibility of other nutrients, but was only a component that diluted the diet.

Adewole [43] noted in his research that extruding oat hull and its use in the amount of 3% of the diet did not significantly deteriorate ($p > 0.05$) feed intake (39.6 g·day$^{-1}$) compared to the control group (40.3 g·day$^{-1}$). The use of non-extruded hull lowered the feed intake by 5.0% in relation to its extruded counterpart, and 6.6% in relation to the broiler chickens' control group. In contrast, the studies conducted by Jimenez-Moreno et al. [44] show that the use of oat hull in the share of 3% resulted in a higher ($p < 0.05$) daily feed intake compared to the control group (41.7 g and 40.1 g, respectively), and in consequence, caused a higher daily weight gain (32, 3 g and 29.3 g) and lower feed conversion per kg ADG (1.29 kg of feed·kg ADG$^{-1}$ vs. 1.36 kg·kg ADG$^{-1}$).

In the conducted studies, the lowest FCR was recorded in the first days of life of the birds receiving 3% of oat hull in their diets. The increased amount of oat hull in the diet of broilers may suggest that in the case of an underdeveloped gizzard, it stimulates the body to absorb more nutrients from digesta. Jimenez-Moreno et al. [44] observed that the use of oat hull or sugar beet pulp in the amount of 3% increased FCR and ADG in younger chickens. On the other hand, Kheravii et al. [45], when examining the use of lignocellulose (0–2%) in diets based on coarse and finely ground maize grain, obtained

similar results when using 2% of this component in diets with coarsely ground maize grain (1.05 kg·kg ADG$^{-1}$). In the case of finely ground maize, the addition of lignocellulose deteriorated the feed conversion. Conversely, Kheravii et al. [46] found no differences in the feed conversion for broiler chickens fed with mixtures containing oat hull or without it up to 35 days of age (1.41 and 1.43 kg of feed·kg ADG$^{-1}$, respectively).

In the current experiment, the worse FRC was noted for diets containing a higher amount of oat hull. However, the differences between the groups with the amount of 1% and 2% of oat hull in diets throughout the experiment were statistically insignificant ($p > 0.05$). Similar results ($p > 0.05$) were obtained by Kheravii et al. [45] in their research using the addition of lignocellulose up to the 24th and 35th days of the birds' lives. In turn, studies by Shirzadegan and Taheri [47] indicate that a change in the carbohydrate source affects the feed utilization of broiler chickens between 25 and 42 days of age. On the other hand, Mulla et al. [42] report that the use of 2% of oat hull in the diet reduced the feed conversion to 1.57 kg of feed·kg ADG$^{-1}$ compared to the control group of birds (1.68 kg of feed·kg ADG$^{-1}$).

*4.2. Morphometric Measurements of the Gastrointestinal Tract*

The weights and lengths of individual segments of the digestive tract largely depend on the content of SDF and IDF and the particle size of components in the diet [43,44,48].

Many studies have shown that the addition of IDF to the diet of broiler chickens affects the development of the upper part of the gastrointestinal tract, in particular, the crop of which the main function is to prepare the ingested food for further digestion [20,23,28,31]. In the conducted experiment, no effect of the type of cereals and the use of oat hull on the weight of the crop in terms of metabolic weight was found. There are not many reports in the available literature on the weight of the crop and the proventriculus in broiler chickens and the possibility of increasing their weight as a result of the use of oat hulls, because these organs do not respond significantly to their addition to the diet. This conclusion was also confirmed in the conducted experiment, as no statistically significant differences ($p > 0.05$) were found in the case of the weight of this organ in terms of metabolic weight. However, significant differences ($p < 0.05$) occurred between chickens receiving corn in comparison with those receiving barley or wheat. A heavier proventriculus was found in birds receiving cereal grains richer in SDF. The exception in terms of the literature are the studies of Amerah et al. [49], who report that the form and degree of the fragmentation of the diet containing wheat grain influence the crop weight of broiler chickens.

In the studies, there was no clear relation in changing the weight of the proventriculus depending on the amount of oat hull used. Similar conclusions were noted by Hetland and Svihus [30], who did not note changes in the weight of this organ between the group of chickens receiving 10% of oat hull in their diet and the control group. In turn, Gonzalez-Alvarado et al. [50] found no differences ($p > 0,05$) in the weight of the glandular stomach between the control group and the birds receiving 3% of its share in their diet. However, they found a significant difference between these groups and the chickens receiving soybean hulls. On the other hand, Taylor and Jones [51] reported a significant reduction in the weight of the proventriculus ($p < 0.01$) in the case of using whole barley or wheat grains in the diet compared to wheat or barley middlings.

A particularly sensitive organ to the size of food particles and the content of IDF is the gizzard, whose main function is to break down food particles. The addition of oat hull to the diet, the use of whole cereal grains or the application of barley grains increase the weight of this organ. This was confirmed in the conducted experiment in the case of chickens receiving barley grain in their diet, as they showed a significantly higher metabolic weight of this organ compared to chickens receiving corn or wheat grain. The weight of the gizzard also increased with the increase of oat hull in their diet as a source of the lignin-cellulose complex. This is also confirmed in the studies of other authors, where the addition of IDF to the diet increases the relative weight of the gizzard and digestive tract of broiler chickens [9,34,52].

Gonzalez-Alvarado et al. [40] also state that the development of the digestive tract of broiler chickens depends on the source of dietary fiber. In these studies, the addition of dried sugar beet pulp was more effective in increasing the weight of the gastrointestinal tract and the amount of digesta in the gizzard, while the use of oat hull mainly affected the weight of the gizzard. Similar conclusions were also reached by Jorgensen et al. [32], who found that the total weight of the digestive tract increases when a soluble source of dietary fiber (pea hull) is added to the diet rather than an insoluble one. Jimenez-Moreno et al. [53] reported a 30% increase in gizzard weight in five-week-old chickens when oat hull was used. In contrast, Gonzalez-Alvarado et al. [50] showed a 35% increase in relative gizzard weight for 21-day-old chicks.

The total length of the intestines in the experiment did not show statistical differences ($p > 0.05$) in the case of the type of cereal grain used. However, in the case of oat hull, no statistical differences were found between the control group and chickens receiving 3% of oat hull in their diet, and 1% of oat hull reduced the total length of the intestines ($p < 0.05$). Amerah et al. [49] in their studies also indicate that the relative length of the intestines is shorter compared with chickens fed diets with whole grain and the control group. In addition, the total length of the intestines is also dependent on the slaughter weight of the birds. Based on the data obtained in the experiment, the main changes in the total length of the intestines resulted from the change in the percentage of duodenal and ileum length. In the case of the duodenum, the presence of barley in the diet of chickens significantly increased its percentage share of the total length of intestines ($p < 0.05$), as well as length in terms of the metabolic weight ($p < 0.01$, Tables 6 and 7), while in the case of the jejunum, the percentage share of the total length of intestines compared to corn ($p < 0.01$) and wheat ($p < 0.05$) by about 0.5 percentage points. With the increase in the content of oat hull in the diet, the length of the duodenum increased in terms of the metabolic weight, but this was not reflected in the percentage share of the duodenum in the total length of the intestines—in this case, the differentiation under the influence of the oat hull amount on the length of the ileum.

### 4.3. Correlation between Development of Gastrointestinal Tract and Microorganisms

In the case of microorganisms in the crop, no significant ($p < 0.05$) relationships between weights and lengths per metabolic weight were observed, with the exception of *E. coli*, the number of which was correlated with an increase in the weight of the proventriculus (weakly, r = 0.392), gizzard (mean, r = 0.486) and duodenal length (strong, r = 0.657), which is related to the production of hydrochloric acid in the proventriculus, the barrier function of the gizzard and duodenogastric reflux [23].

The metabolic weight of the glandular stomach was mainly influenced by the contribution of WS, TDF and ADF in the diet (r = 0.586; r = 0.502; r = 0.552, respectively). The use of 2–3% IDF in the diet stimulates the secretion of enzymes in the proventriculus and duodenal lumen and influences the development of the muscular stomach [54]. In this case, the higher weight of this organ, and thus, the possible greater area of pepsinogen and HCl production, was additionally due to the reduction of protein absorption in the small intestine of birds receiving mixtures with a 50% share of wheat or barley grain (3.4 and 3.6 g, respectively) compared to a corn diet (3.3 g). Reazei and Hajati [27], in studies with the use of rice hull, did not show significant changes in the weight of this organ expressed as a percentage of the carcass weight in broiler chickens when it was used in a share of up to 40% for a short period of 16 to 20 days between the transition from grower to the finisher diet.

The metabolic weight of the proventriculus was mainly influenced by the contribution of CF, TDF and ADF in the diet (r = 0.586; r = 0.502; r = 0.552, respectively). The use of 2–3% IDF in the diet stimulates the secretion of enzymes in the proventriculus and duodenal lumen and also influences the development of the gizzard [54]. In this case, the higher weight of this organ, and thus, the possible greater area of pepsinogen and HCl production, was additionally due to the reduction of protein absorption in the small intestine of birds

receiving diets with a 50% share of wheat or barley grain (3.4 and 3.6 g, respectively) compared to a corn diet (3.3 g). Reazei and Hajati [24], in studies with the use of rice hull, did not show significant changes in the weight of this organ expressed as a percentage of the carcass weight in broiler chickens when it was used in the amount of up to 40% for a short period of 16 to 20 days between the transition from grower to the finisher diet.

Jimenez-Moreno et al. [31], based on the research of, inter alia, Hetland et al. [55], noted that the content of IDF in the diet changes the size and weight of the gizzard, which plays an important role in the fragmentation of feed particles and regulates the flow of digesta to the small intestine. Changes in the digesta flow rate were also observed by Kim et al. [56].

Research by Mateos et al. [5] also indicates the fact of longer retention of digesta in the gizzard in the case of an increased amount of insoluble fiber in the diet. The results of the experiment show that the application of a component that is an additional source of IDF in the diet of broiler chickens is positively correlated with the weight of the muscular stomach. This is confirmed by the results of the experiment by Razei and Hajati [27], in which oat hull in 3 experimental groups (0, 20 and 40% of the concentrated mixture on days 16 to 20) was used, where an increase in the percentage of the gizzard weight in the carcass weight was noted ($p < 0.05$) and it increased from 1.30% to 1.44% (respectively for 0 and 40% of rice hull in the diet). This is also confirmed by the positive correlation between the gizzard weight and the crude fiber, TDF, IDF and ADF content (strong correlations, r = 0.733; r = 0.662; r = 0.607; r = 0.785, respectively).

A consequence of the increased amount of IDF (r = 0.533) and ADF (r = 0.575) in the diet is also an increase in the length of the duodenum recalculated on metabolic weight ($p < 0.01$) in the case of barley grain diets to 8.8 g, while for maize and wheat grains it was 8.2 and 8.4 g, respectively. Sabour et al. [57] also noted the increase in the length of the microvilli of this section in response to the increased amount of insoluble fiber (rice hull) in the diet compared to the control group fed a standard diet.

When analyzing the use of oat hull, no significant statistical differences were observed, except for the ileum, where after the use of 1% of oat hull in the diet, its length was shortened compared to the 3% share ($p < 0.05$) and the lack of this structural component ($p < 0.01$). This may be suggested by the fact that a slight increase in the amount of IDF in the feed increases the digestibility of starch and protein within the ileum [9,19].

When converting the length of the intestines as a percentage of the total length, it was observed that the shortening of the total length of the intestines ($p < 0.01$) occurred when the animals were fed with feed containing 1% of oat hull in their diet, compared to 0 or 3% of this component in the feed. Han et al. [20], in a study of ducklings up to 21 days of age, found no significant differences in the relative intestinal weight of the duodenum, jejunum, ileum and single cecum with the use of identical dietary levels of wheat and rice hull (from 0 to 30% of the content of the diet). On the other hand, with regard to the ileum, an average positive correlation was found between the length of the jejunum and TAMC (r = 0.407). The increase in duodenal length caused a strong reduction in the count of *E. coli* bacteria (r = −0.593), which may be related to the secretion of a higher amount of HCl by the proventriculus and the barrier effect of the gizzard in limiting these bacteria in the ileum.

## 5. Conclusions

- Barley grain and oat hull in the amount of 3% increased the metabolic weight of the gizzard.
- The use of 1% of oat hull in the diet shortened the total length of the intestines and the highest body weight of the birds was recorded in this group.
- SDF contained in barley and wheat grain resulted in a higher metabolic weight of the proventriculus of broiler chickens compared to birds receiving corn grain in their diet ($p \leq 0.05$).

- The increase in the weight of the glandular stomach (r = 0.392), muscle (r = 0.486) and duodenum (r = 0.657) was positively correlated with the count of *E. coli* bacteria in the crop indicating a barrier effect, especially of the gizzard.
- The increase in the content of the crude fiber, IDF and ADF contained in diets was positively correlated in broiler chickens with an increase in the weight of the proventriculus, gizzard and duodenal length.
- The increase in the weight of the proventriculus (r = 0.392), gizzard (r = 0.486) and duodenum (r = 0.657) was positively correlated with the growth of *E. coli* bacteria in the crop. The opposite effect in the case of negative correlation was determined in the case of the duodenum and *E. coli* count (r = −0.593).

**Author Contributions:** Conceptualization, T.H. and A.W.; methodology, T.H. and A.W.; formal analysis, T.H., P.W. and A.S.-T.; investigation, P.W. and A.S.-T.; resources, T.H.; data curation, T.H.; writing—original draft preparation, T.H., P.W. and A.S.-T.; writing—review and editing, T.H., P.W. and A.S.-T.; visualization, T.H. and A.S.-T.; supervision, A.W.; funding acquisition, T.H. All authors have read and agreed to the published version of the manuscript.

**Funding:** Polish National Science Centre pre-doctoral grant, N311 476239.

**Institutional Review Board Statement:** Not applicable.

**Data Availability Statement:** The data presented in this study are available on request from the corresponding author.

**Conflicts of Interest:** The authors declare no conflict of interest.

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
