# Peer review of "Oat Hull as a Source of Lignin-Cellulose Complex in Diets Containing Wheat or Barley and Its Effect on Performance and Morphometric Measurements of Gastrointestinal Tract in Broiler Chickens"

_agriculture, doi:10.3390/agriculture13040896_

Round 1

Reviewer 1 Report

  1. In Table 7 units of „Slaughtery weight“ are missing.

2. There is no negative control group in this study. How the effect of treatments could be explained, if there isn‘t presented the group without treatment?

3.   Please expand information in introduction or material and methods, why broiler chickens were reared only till 28th day of age? Normaly, the chickens are kept till 35 or 40 days of age.

4.       In results chapter the data in text could be expressed in %, but not in grams, kg or cm. These units are presented in tables, so in text the change of treatments could be expressed in %. It would be more informative, especially in body weight or FCR data.

Reviewer 2 Report

The research is based on the novel idea to explore the effect of dietary, especially soluble and insoluble fiber from oat hulls on the performance, gut and organs weight of the broiler. 

The manuscript is written well. A t few places, I found spelling or sentences clarity issues. Please correct it. Comments are on the manuscript file. 

Reviewer 3 Report

see attachment!

Reviewer 4 Report

After a careful review of the manuscript entitled “Effect of oat hull in diets containing wheat and barley on per- 2 formance, morphometric measurements of the gastrointestinal 3 tract and its correlation with microorganisms and fraction of 4 the dietary fibre in broiler chickens” here are few minor general recommendations. Overall, it is a well written and informative article, but I suggest some revisions in the article. Please address the following points in the revised manuscript.

1.      Please add one are more sentences about the novelty in the abstract.

2.      Objectives of the study should be clearly mentioned in the manuscript with more details.

3.      In the abstract, add a conclusive line. 

4. Rational is missing in the manuscript 

Round 2

Reviewer 1 Report

There is no additional comments for authors.

Author Response

Thank you for your comment.

Reviewer 3 Report

Good Paper

Author Response

Thank you for your comment.